# Low-salinity medium for large-scale biomass production of the marine purple photosynthetic bacterium *Rhodovulum sulfidophilum*

Shamitha Rao Morey-Yagi[1,3]*, Dao Duy Hanh[1], Miki Suzuki[1], Shota Kato[2], Geoffrey Liou[2], Yuki Kuroishikawa[2], Ayaka Yamaguchi[2], Hiromasa Morishita[4], Masaki Odahara[3,1¤a], Keiji Numata[1,2,3]*

1 Laboratory for Biomaterial Chemistry, Department of Material Chemistry, Graduate School of Engineering, Kyoto University, Nishikyo-ku, Kyoto, 615-8246, Japan, 2 Symbiobe Inc., Venture Plaza South Building, Nishikyo-ku, Kyoto, 615-8245, Japan, 3 Biomacromolecules Research Team, RIKEN Center for Sustainable Resource Science, 2-1 Hirosawa, Wako, Saitama, 351-0198, Japan, 4 Support Unit for Bio-Material Analysis, RIKEN Center for Brain Science, 2-1 Hirosawa, Wako, Saitama, 351-0198, Japan.

¤a Current address
* yagi.shamitharao.2a@kyoto-u.ac.jp (SRM-Y); keiji.numata@riken.jp (KN)

## Abstract

Marine purple non-sulfur bacteria such as *Rhodovulum sulfidophilum* are versatile due to their diverse applications in bioremediation, biotechnological production of useful materials, industrial production of value-added compounds, and agricultural fertilizers. Our previous study demonstrated the potential of its lysed and dried biomass as a nitrogen fertilizer for plant production. However, large-scale fertilizer production requires scaling up the culture to larger volumes, which is not cost-effective with currently available options for growth media. In this study, we tested a seawater-based, cost-effective alternative to the commonly used nutrient-rich culture medium for the growth of this bacterium. We found that reducing salinity from 3% to 1.2% had no adverse effects on its heterotrophic growth, dry cell yield, nitrogen content, and total amino acid composition. The nitrogen content and the weight percent of free lysine, aspartic acid, and glutamate tended to increase in the biomass obtained from cultures grown at 1.2% salinity. Under autotrophic conditions, decreasing salinity to 1.2% did not affect cell growth, final dry cell yields, and total carbon assimilation, but N assimilation remained higher. Reducing salinity to 1.2% proved to be cost-effective and feasible for the cultivation of *R. sulfidophilum* without increasing the risk of contamination, providing a viable alternative for its large-scale cultivation and application as a plant nitrogen fertilizer.

**Data availability statement:** All relevant data are within the manuscript and its Supporting Information files.

**Funding:** This work was supported by Japan Science and Technology, COI-Next (Grant Number JPMJPF2114).

**Competing interests:** Numata K., Morey-Yagi SR., and Symbiobe Inc. are co-inventors of a patent for fertilizer produced using R. sulfidophilum biomass (Patent pending JP2022-076662). Kato S., Liou G., Kuroishikawa Y., Yamaguchi, A., and Numata K. have affiliations with Symbiobe Inc. This does not alter our adherence to PLOS ONE policies on sharing data and materials.

## Introduction

Marine purple non-sulfur bacteria (PNSB) are photomixotrophs capable of using both $CO_2$ (autotrophy) and organic carbon (C) compounds (heterotrophy) to meet their C needs. Many PNSB species can also acquire nitrogen (N) by fixing $N_2$ using endogenous nitrogenases [1,2]. Additionally, they can produce value-added compounds such as carotenoids [3,4], vitamins [4,5], and polyhydroxyalkanoates [6–10]. These bacteria are reported to tolerate a wide range of salinity [4,11,12] and are predominantly grown in culture medium with 2% – 3% NaCl [7,9,10,12–15]. The use of natural seawater (approximately 3% – 3.5% salinity) for their culture has been previously proposed for the sustainable, cost-effective production of these value-added compounds with a low risk of biological contamination [13,16,17]. However, this has not been tested yet.

Our earlier studies demonstrated that a gram-negative marine PNSB, *Rhodovulum sulfidophilum* (ATCC 35886), can synthesize polyhydroxyalkanoates (PHA) under photoautotrophic [7–10,17] and heterotrophic [13,15] conditions. It can also be used as a host for the heterologous expression of spider silk [14]. Additionally, we have reported that its lysed and dried biomass can be used as a sustainable N fertilizer for plant cultivation due to its N content of ~11% and protein content of ~69% [18]. This allows biomass waste to be repurposed as plant fertilizer after the extraction of value-added compounds. However, large-scale cultivation of this bacterium in a nutrient-rich medium towards this end is not without significant costs. Therefore, in this study, we evaluated the growth, nutrient (N, P, K), and amino acid profile of *R. sulfidophilum* in a low-salinity medium, intending to improve biomass yield and reduce large-scale culture costs, without additional risks of contamination.

## Results

### An artificial seawater-based culture medium is cost-effective for the heterotrophic cultivation of *R. sulfidophilum*

We compared the cell yield and culture media costs of culturing *R. sulfidophilum* in three different growth media namely, commercially available marine broth (MB), a natural seawater (NSW) based and an artificial seawater (ASW) based medium both supplemented with 0.1% yeast extract and 0.5% peptone in a 10 L scale. The fresh cell yield in MB was significantly higher than the seawater-based media, but their dry cell yields were similar (Table 1 and S1 Table), indicating that the seawater-based media could replace MB. Using NSW- and ASW-based culture media instead of MB decreased the culture media costs by 90% and 91%, respectively (Table 1). Additionally, culturing in the ASW-based medium was 9% cheaper than using the NSW-based culture medium (Table 1). Hence, we used the ASW-based culture medium for further salinity tests.

### Lower salinity has no negative effects on the heterotrophic growth of *R. sulfidophilum* in ASW-based media

*R. sulfidophilum* was cultured in 15 mL of ASW supplemented with 0.1% yeast extract and 0.5% peptone (Figs 1a and 1b) or 1 mM sodium acetate and 2.5 mM sodium

**Table 1. Cell yield and cost analysis of various growth media for large-scale cultivation of *Rhodovulum sulfidophilum*.**

| | Fresh cell yield (g L$^{-1}$) | Dry cell yield (g L$^{-1}$) | Cost for 1 L (yen) | Cost relative to MB | Cost relative to NSW |
|---|---|---|---|---|---|
| MB (76448, Millipore) | 6.99±0.76[a] | 0.96±0.10 | 1541 | | |
| NSW+0.1% YE+0.5% Peptone | 4.65±1.10[b] | 1.03±0.23 | 148 | −90% | |
| ASW+0.1% YE+0.5% Peptone | 5.03±0.68[b] | 0.88±0.10 | 134 | −91% | −9% |

Data represent means±standard error of the means (SEM) from 4 independent 10 L cultures, or batches (n=4). Different letters indicate the significance of differences observed with Marine Broth (MB) as evaluated by a one-way ANOVA (Dunnett's test) (GraphPad Prism 9).

thiosulfate (Fig 1c) at decreasing concentrations of ASW, i.e., 100%, 90%, 80%, 70%, 60%, and 50% which correspond to 3%, 2.7%, 2.4%, 2.1%, 1.8%, and 1.5% salinities respectively. The optical density (OD) of cultures at 660 nm in 60% ASW and 50% ASW was higher than in 100% ASW at 72, 128, and 177 hours and 128 and 177 hours of culture, respectively (Fig 1a, S2 and S3 Tables). But there were no significant effects of lower salinities (down to 1.5%) on dry cell yield (Fig 1b and S4 Table). Similar results were also obtained for cultures supplemented with 1 mM sodium acetate and 2.5 mM sodium thiosulfate (Fig 1c and S5 Table), wherein a decrease in salinity had no negative impact on the dry cell yield.

### *R. sulfidophilum* shows comparable growth, nutrient, and amino acid profile in 40% ASW-based heterotrophic culture medium (1.2% salinity) without additional contamination risks

To evaluate the impact of lower salinities (50%, 40%, and 30% ASW corresponding to 1.5%, 1.2%, and 0.9% salinities respectively) on large-scale growth of *R. sulfidophilum*, it was cultured in ASW supplemented with 0.1% yeast extract and 0.5% peptone in 10 L bottles (Fig 2a). The dry cell yields tended to be higher in 50% and 40% ASW, but lower in 30% ASW compared to 100% ASW (Fig 2b and S6 Table). Thus, 40% ASW or 1.2% salinity was used as the low salinity medium for testing growth, nitrogen: phosphorus: potassium (N: P: K) and the amino acid profile in 10 L scale to evaluate the implications of this reduction on its use as a plant N fertilizer. The initial and final OD$_{660}$ (Fig 2c and S7 Table) and the initial and final dry cell yield (Fig 2d and S8 Table) were not different between the 100% and the 40% ASW treatments, but tended to be higher in the lower salinity medium. Further, there were no additional risks of contamination caused by lowering the salinity to 1.2% (Fig 2e). The biomass obtained from the respective treatments was lysed and dried (Processed Biomass or PB hereafter) for further analysis. The N: P: K of PB from 40% ASW treatment was 11.9: 2.94: 0.57, comparable to previously reported values in MB (11.0: 2.95: 0.51) [18], with a slight tendency toward higher N in the former. The total amino acid composition between the PB from 100% and 40% ASW was comparable, except for glutamate and glutamine, which tended to be higher in 40% ASW (Table 2). The composition of free amino acids, on the other hand, was marginally different, with a trend towards higher lysine, glutamate, and aspartate; and lower arginine in PB from 40% ASW (Table 2). Additionally, decreasing the salinity of the ASW-based culture medium by 60% cut culture costs by 92% and 12% relative to MB and NSW-based culture medium, respectively.

### 40% ASW can replace 100% ASW for the autotrophic growth of *R. sulfidophilum*

To test if 40% ASW could replace the 100% ASW in autotrophic conditions, *R. sulfidophilum* was cultured in 100% and 40% ASW supplemented with 10 mM sodium thiosulfate pentahydrate and a CO$_2$/N$_2$ (7:3) gas mixture for 4 days in 2 L jar fermenters (Fig 3a). Cells grew similarly (based on OD$_{660}$) between the two treatments (Fig 3b and S9 Table), and a comparable final dry cell yield of ~0.4 g L$^{-1}$ was obtained for the two treatments (Fig 3c and S10 Table). However, 40% ASW, i.e., the 1.2% salinity condition, tended to display higher initial growth than 100% ASW, i.e., the 3% salinity condition (Figs 3b and 3c). Specifically, the growth rate during the first day of culture was higher at 1.2% salinity, while the growth rates on day 2 and day 3 remained higher at 3% salinity (Fig 3d and S9 Table). Similar trends were also observed with the total organic carbon (TOC) and the total nitrogen (TN) assimilated by cells, which were obtained from the sole C and

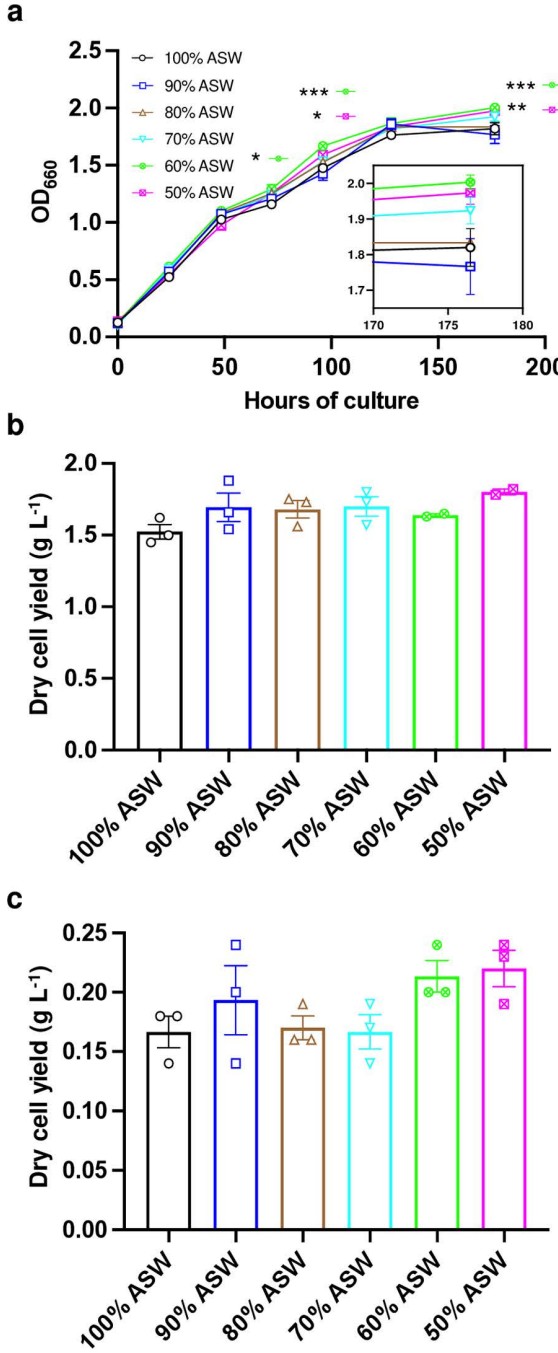

**Fig 1. Heterotrophic cell growth and dry cell yield of *Rhodovulum sulfidophilum* in artificial seawater (ASW) based growth medium of different salinities, wherein 100% corresponds to 3% salinity.** **a,** Growth curve and **b,** dry cell yield (g L$^{-1}$) of *R. sulfidophilum* in ASW supplemented with 0.1% yeast extract and 0.5% peptone, and **c,** dry cell yield (g L$^{-1}$) of *R. sulfidophilum* in ASW supplemented with 1 mM sodium acetate and 2.5 mM sodium thiosulfate in decreasing concentrations of ASW, i.e., 100%, 90%, 80%, 70%, 60%, and 50% which correspond to 3%, 2.7%, 2.4%, 2.1%, 1.8%, and 1.5% salinities respectively. Zoomed inlay in Fig 1a shows OD at 660 nm for each treatment at 176.5 hours. Data represent means ± standard error of the means (SEM) for 3 independent 15 mL batch cultures (n = 3). Asterisks indicate significant differences between 100% and decreasing concentrations of ASW tested using one-way ANOVA (Figs 1b and 1c) or two-way ANOVA (Fig 1a), and the Dunnett's multiple comparison test (GraphPad Prism 9); * p < 0.05, ** p < 0.01, and *** p < 0.001.

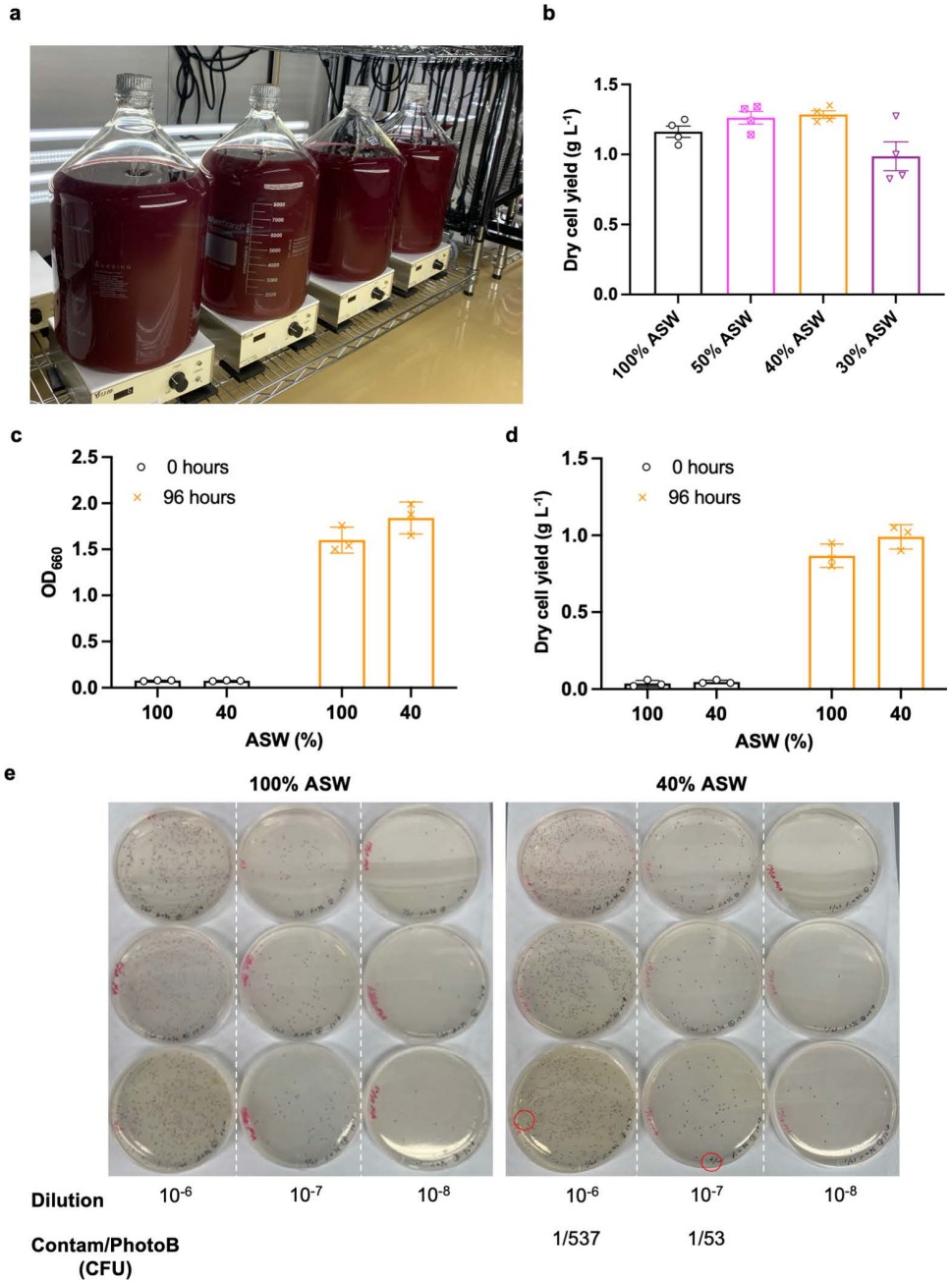

**Fig 2. Dry cell yields and contamination status in ASW-based *R. sulfidophilum* cultures at a reduced salinity of 1.2%; 100% and 40% correspond to 3% and 1.2% salinity respectively.** **a,** Ten-liter culture set-up and **b,** dry cell yield (g L$^{-1}$) of *R. sulfidophilum* in ASW supplemented with 0.1% yeast extract and 0.5% peptone at 100%, 50%, 40%, and 30% ASW, corresponding to 3%, 1.5%, 1.2%, and 0.9% salinities respectively. **c,** Initial and final OD at 660 nm, and **d,** the dry cell yield (g L$^{-1}$) was obtained for 100% and 40% ASW treatments. **e,** Contamination status at the end of the culture period was obtained for 10$^{-6}$ to 10$^{-8}$ serial dilutions of 100% (*left panel*) and 40% (*right panel*) ASW treatments. Data represent means ± SEM for n = 3 or 4. Significant differences between all salinities tested (Fig 2b), and the 3% and 1.2% salinity treatments (Figs 2c and 2d) were not observed using a one-way ANOVA (Dunnett's test) (GraphPad Prism 9) or Student's T-test statistic (Microsoft Excel 2019), respectively. Red circles indicate the contaminating colonies.

**Table 2. Free and total amino acid profile expressed in weight % of PB from *R. sulfidophilum* cultured in 100% and 40% ASW.**

| Amino acid | Total amino acid content (weight %) | | Free amino acid content (weight %) | |
|---|---|---|---|---|
| | 100% ASW | 40% ASW | 100% ASW | 40% ASW |
| Arg | 7.62 | 7.42 | 37.56 | 30.28 |
| Asp+Asn | 10.65 | 10.47 | 14.11 | 15.77 |
| Glu+Gln | 13.73 | 14.54 | 12.43 | 14.68 |
| Lys | 5.40 | 5.54 | 16.33 | 20.42 |
| Ser | 3.54 | 3.41 | 0.29 | 0.12 |
| Thr | 4.77 | 4.76 | 0.50 | 0.45 |
| His | 1.87 | 1.84 | 0.00 | 0.00 |
| Met | 3.11 | 3.11 | 0.64 | 0.62 |
| Ala | 9.20 | 9.05 | 1.56 | 1.66 |
| Val | 6.84 | 6.80 | 1.74 | 1.33 |
| Gly | 5.86 | 5.82 | 3.09 | 3.02 |
| lle | 4.83 | 4.77 | 1.93 | 2.14 |
| Leu | 9.43 | 9.32 | 3.96 | 3.37 |
| Phe | 5.36 | 5.36 | 3.05 | 3.18 |
| Pro | 4.25 | 4.19 | 1.00 | 1.24 |
| Trp | n.d. | n.d. | 0.41 | 0.43 |
| Tyr | 3.53 | 3.60 | 1.21 | 1.10 |

Data are shown for the combined PB from 3 independent batch cultures; error bars not shown as statistical variation could not be calculated from pooled data. Values should be interpreted as representative trends. n.d. indicates no data due to degradation by HCl.

N sources, $CO_2$ and $N_2$, respectively. TOC of cells in 1.2% salinity was higher during initial growth (Fig 3e and S11 Table), consistent with cell growth (Figs 3b and 3c). However, the TN remained higher in cells throughout the culture in 1.2% salinity compared to 3% salinity (Fig 3f and S11 Table). Thus, 1.2% salinity was more effective than 3% salinity in supporting autotrophic growth of *R. sulfidophilum* in ASW-based growth medium.

## Discussion

We evaluated the performance of *R. sulfidophilum* in a low-salinity growth medium as a cost-effective alternative to MB and confirmed its effectiveness as a plant N fertilizer. Lowering the salinity to 1.2% did not adversely affect cell growth and the N, P, K content, but tended to improve dry cell yield (Figs 2d and 3c), N content, and alter the amino acid composition (Table 2). While one of the advantages of using natural seawater (salinity of ~3% or above) as a culture medium is its low risk of contamination, decreasing the salinity of seawater-based medium to 1.2% did not increase the contamination events during cultivation in 10 L-scale (Fig 2e). Thus, the culture medium for large-scale cultivation of *R. sulfidophilum* was comprehensively improved without any trade-offs. However, this study primarily focused on short-term growth responses under reduced salinity conditions. To assess the feasibility of this approach for sustained industrial use, further evaluation of long-term culture stability and viability across multiple subculturing cycles may be necessary.

The trend of higher growth (Fig 1a) and dry cell yield in culture medium with lower salinities (Figs 1b and 1c) is consistent with a previous study, wherein decreased PHA accumulation in cells was also observed with a decrease in salinity from 4.5% to 1.5% [19]. Acetyl-CoA is the precursor for PHA biosynthesis in *R. sulfidophilum* [9,15] and also a common metabolic intermediate of the C metabolism feeding into the TCA cycle, which provides C skeletons necessary for N metabolism, including the biosynthesis of amino acids [20]. Hence, a decrease in C flux towards PHA biosynthesis could increase the flux towards the TCA cycle, facilitating improved growth and/or N assimilation in the lower salinity growth medium. This was also observed in the current study, wherein cells when grown autotrophically, tended to assimilate more

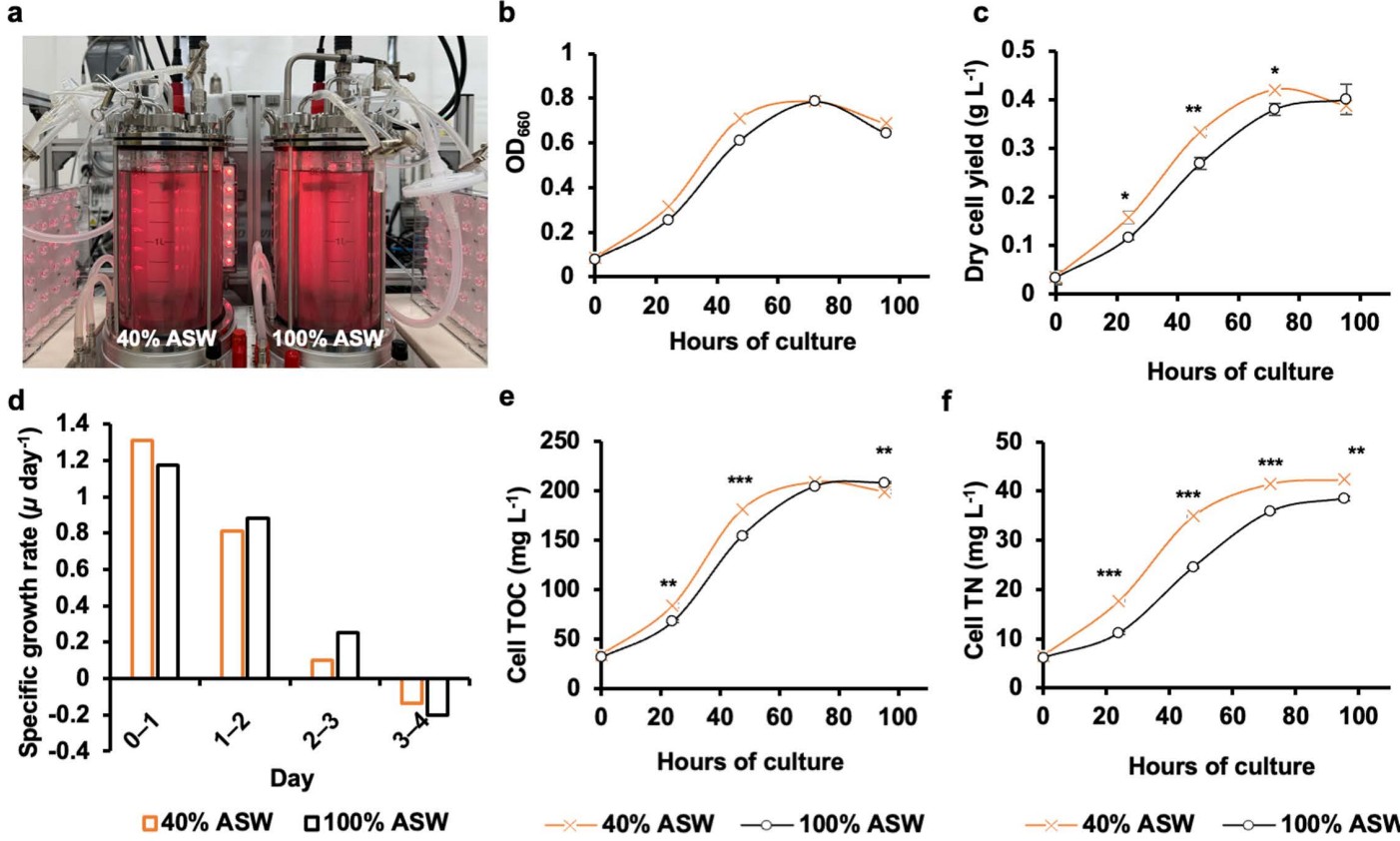

**Fig 3. Cell growth, dry cell yield, total organic carbon (TOC), and total nitrogen (TN) in autotrophic cultures of *R. sulfidophilum* in artificial seawater (ASW) based growth medium of different salinities, wherein 100% corresponds to 3% salinity and 40% corresponds to 1.2% salinity. a,** Two-liter fermenter set-up for the culture of *R. sulfidophilum* in 100% or 40% ASW supplemented with 10 mM sodium thiosulfate pentahydrate and $CO_2/N_2$ (7:3) gas mixture. **b,** OD at 660 nm, **c,** dry cell yield (g $L^{-1}$), and **d,** OD-based specific growth rates ($\mu$) were obtained for every day of the culture. The total C and N in the cell fractions are shown as **e,** cell TOC (mg $L^{-1}$), and **f,** cell TN (mg $L^{-1}$), respectively. Data represent means ± SEM from three technical replicates (n = 3). Asterisks indicate significant differences between the 3% and 1.2% salinity treatments using the Student's T-test statistic (Microsoft Excel 2019); * $p < 0.05$, ** $p < 0.01$, and *** $p < 0.001$.

N in the 1.2% salinity growth medium (Fig 3f). A similar tendency may also be expected in heterotrophic cultivation leading to marginally higher N content (11.9%) in PB at 1.2% salinity. However, due to the difference in metabolic modes, further study would be necessary to verify the mechanism of a higher N utilization and N content in a 1.2% salinity medium under respective growth conditions.

Previously, we have shown that PB from *R.sulfidophilum* cultures in MB (N: P: K = 11.0: 2.95: 0.51) can be effectively used as N fertilizer for the cultivation of Japanese mustard spinach (komatsuna) [18]. In this study, PB obtained from 1.2% salinity medium displayed comparable N: P: K of 11.9: 2.94: 0.57, indicating its feasibility for use as a plant N fertilizer. Furthermore, PB from a 1.2% salinity growth medium tended to display higher free lysine and glutamate (Table 2), which are reported to play a role in the induction of stress responses in tomato [21] and rice [22], respectively. Free aspartic acid, exogenous application of which has been reported to mitigate the adverse effects of salt stress in wheat [23], also tended to be higher in a 1.2% salinity growth medium. However, free arginine, which plays a minimal role in stress resilience but is a good N source [24,25], tended to be lower in PB from 1.2% salinity growth medium. Although these findings point towards the possible function of PB in enhancing stress resilience in plants, further experiments would be required to confirm its simultaneous use as a fertilizer and a biostimulant.

Finally, the use of low-salinity ASW-based growth medium for large-scale cultivation offers environmental advantages over conventional nutrient-rich formulations. These include reduced nutrient input and discharge, which lower the risk of eutrophication in vulnerable aquatic systems, as well as decrease salinity in waste streams, thereby simplifying down-stream treatment and disposal. Leveraging the autotrophic growth of *R. sulfidophilum* could further reduce dependency on organic inputs, lowering the demand on land, water, and animal-derived byproducts, thus decreasing the carbon and energy footprint associated with biomass production.

## Methods

### Cultivation of *Rhodovulum sulfidophilum*

*Rhodovulum sulfidophilum* (DSM 1374; equivalent to LMG 5202 obtained from ATCC) was used for all experiments in this study, and cultivated in various media with different salt compositions as detailed below. The cultures were maintained at room temperature and 80 W m$^{-2}$ (Y104-R660/W40/IR73-31W-EI0U1LW-010, YUMEX Solutions) with a starting OD$_{660}$ of 0.1.

*Seawater based medium:* Media ranging from 50% (15 g L$^{-1}$) to 100% (30 g L$^{-1}$) ASW (Marine standard, Nihon Kaisui Co. Ltd, Tokyo, Japan) in 10% increments were supplemented with either 0.1% (w/v) yeast extract and 0.5% (w/v) peptone or one mM sodium acetate and 2.5 mM sodium thiosulfate, and used to culture *R. sulfidophilum* to obtain the growth curve and cell dry weights. NSW harvested from Maizuru Bay was passed through 5 μm and 0.5 μm polypropylene wound cartridge filters (Advantec, Tokyo, Japan) into 6 UV sterilizers (UV40W, Kankyo Technos Co. Ltd., Wakayama, Japan) connected in tandem, and circulated through this system periodically to prevent contamination during storage. The cultures in NSW supplemented with 0.1% (w/v) yeast extract and 0.5% (w/v) peptone were used for cell yield and cost analysis comparisons with MB and ASW-based medium. All media were sterilized in an autoclave at 121°C at 15 psi for 30 minutes (10 L scale). Cultures in 100% and 40% ASW supplemented with 0.1% (w/v) yeast extract and 0.5% (w/v) peptone in a 10 L culture scale were used for nutrient (N, P, K) and amino acid analysis.

### Growth analysis and yield of dry cell weight

The OD$_{660}$ readings were taken every 24 hours to plot growth curves for three independent biological replicates under each salinity condition. A fixed volume of 7-day-old (Fig 1) or 4-day-old cell cultures (Figs 2 and 3) were centrifuged at 9000 x*g* for 10 minutes, washed with 1 ml sterile water, transferred to pre-weighed empty 1.5 ml centrifuge tubes, and centrifuged again at 9000 x*g* for 20 minutes. The pellets were lyophilized to determine cell dry weights (g), which were then converted to yield (g L$^{-1}$) based on the volume of the culture used for centrifugation.

### Contamination check, harvest, lysis and drying

Contamination was checked by plating 100 μl of 10$^{-6}$, 10$^{-7}$ and 10$^{-8}$ serial dilutions of 4-day-old cultures on marine agar [Marine Broth (Merck, Millipore Sigma, Massachusetts, United States) with 1.5% Agar], and incubating at 30°C for 3 days under far-red light (25 W m$^{-2}$). Colony forming units (CFU) of *R. sulfidophilum* and contaminants were counted on plates showing contamination.

Cells from 4-day-old 10 L cultures were first flocculated in 7.5 mg L$^{-1}$ chitosan 100 (Fujifilm Wako, Osaka, Japan) in 0.01% acetic acid (Fujifilm Wako) for 2–3 days, which were then harvested by centrifugation (14,000 x*g*, 10 minutes), and resuspended in pure water to remove the salt. For cells cultured in marine broth, 12 mg L$^{-1}$ of chitosan 100 in 0.01% acetic acid was used for flocculation. Washed cells were then collected by centrifugation (14,000 x*g*, 60 minutes) and stored at –80°C until further use. Cell pellets from 3 independent culture bottles in each salinity condition were combined, chemically lysed, air-dried, and powdered for nutrient and amino acid analysis.

## Nutrient and amino acid analysis

PB was analyzed for N and C content by dry combustion. Phosphate ($P_2O_5$) and potassium oxide ($K_2O$) were extracted through nitric acid decomposition and analyzed using absorptiometry and atomic absorption spectroscopy, respectively, at Vegetech Co., Ltd. Physical and Chemical Analysis Center, Kanagawa, Japan. Values are expressed as % dry weight.

The amount of free (not bound in a protein) and total (proteinogenic + free) amino acids (except for cysteine and tryptophan) in PB were quantified with an amino acid analyzer (L-8900, Hitachi) using the post-column ninhydrin derivatization of amino acids. For the quantification of free amino acids, 5 mg of the air-dried sample was subjected to sonication in 200 µL of 0.2 M perchloric acid for ~2 minutes and incubated at 0°C for 30 min. Following centrifugation at 12,000 x$g$ for 5 minutes, the supernatant was diluted 3.3-fold with lithium citrate buffer (pH 2.2) (123–02505, Fujifilm Wako Pure Chemical Corporation), and 90 µL was used for the analysis. For the quantification of total amino acids (18 of the 20 proteinogenic amino acids, excluding cysteine and tryptophan), 1 mg of the lyophilized sample was subject to hydrolysis in 100 µL of 6 M hydrochloric acid at 110°C for 20 hours. The volume of the hydrolysate was adjusted to 100 µL with ultra-pure water and filtered through a 0.45 µm filter. The filtrate was diluted 20-fold with the lithium citrate buffer and 15 µL was used for the analysis.

## Autotrophic cultivation and Total organic Carbon (TOC)/ Total Nitrogen (TN) analysis

*R. sulfidophilum* was autotrophically cultured in 100% and 40% ASW supplemented with 10 mM sodium thiosulfate pentahydrate and a $CO_2/N_2$ (7:3) gas mixture in two independent 2 L jar fermenters (BEM, Tokyo, Japan) at 27°C for 4 days under far-red light (25 W m$^{-2}$). The culture medium was bubbled with $N_2$ for 15 minutes at 0.5 L min$^{-1}$ to purge the dissolved oxygen. The gas mixture was then bubbled through the culture medium for 30 minutes at 0.5 L min$^{-1}$, and the pH of the culture was maintained at 7.5 using 1N NaOH. The gas exhaust line was then sealed to establish a closed system. Following the bubbling, sodium thiosulfate pentahydrate and the seed culture of *R. sulfidophilum* cultured in M6 minimal medium, as previously reported [10], were inoculated into the culture medium to a final $OD_{660}$ of 0.1. Samples were collected at the start of the culture (0 hours) and 24 hours, 47.5 hours, 72 hours, and 95.5 hours of culture to evaluate growth ($OD_{660}$), dry cell yield, total organic carbon (TOC), and total nitrogen (TN) content. TOC and TN were obtained for ~43 ml of culture and the culture medium, and the samples for culture medium at corresponding time points were prepared by centrifugation of cultures at 9000 x$g$ for 10 minutes. TOC and TN were analyzed by TOC-L and TNM-L (Shimadzu, Kyoto, Japan). TOC was obtained by subtracting the inorganic C from the total C present in the culture or the culture medium (supernatant) at a given time point. The amounts of C and N fixed by the bacteria were calculated by subtracting the amounts of TOC or TN in the culture medium (supernatant) from the TOC or TN in the culture at each corresponding time point. The cells were harvested and processed for the N, P, K analysis as mentioned in the earlier sections.

## Supporting information

**S1 Table. Values behind means, standard error of the means (SEM), and *p*-values used in Table 1. Cell yield from large-scale cultivation of *Rhodovulum sulfidophilum* in various growth media.** Data are presented for four independent 10 L batch cultures (n = 4). *P* values were obtained from one-way ANOVA (Dunnett's test) (GraphPad Prism 9) by comparing marine broth (MB) with natural seawater (NSW) and artificial seawater (ASW) based media.
(PDF)

**S2 Table. Values behind means and SEM used in Fig 1a** . Data used for the growth curve ($OD_{660}$) of *R. sulfidophilum* in ASW supplemented with 0.1% yeast extract and 0.5% peptone in decreasing concentrations of ASW, i.e., 100%, 90%, 80%, 70%, 60%, and 50% which correspond to 3%, 2.7%, 2.4%, 2.1%, 1.8%, and 1.5% salinities respectively (Fig 1a). Data are presented for three independent 15 mL batch cultures (n = 3).
(PDF)

**S3 Table. *p*-values used in Fig 1a** *p* values from two-way ANOVA (Dunnett's test) (GraphPad Prism 9) of $OD_{660}$ comparing 100% with decreasing concentrations of ASW in <ins>Fig 1a</ins>.
(PDF)

**S4 Table. Values behind means, SEM, and *p*-values used in Fig 1b.** Dry cell yield (g $L^{-1}$) of *R. sulfidophilum* in ASW supplemented with 0.1% yeast extract and 0.5% peptone in decreasing concentrations of ASW, i.e., 100%, 90%, 80%, 70%, 60%, and 50% which correspond to 3%, 2.7%, 2.4%, 2.1%, 1.8%, and 1.5% salinities respectively (Fig 1b). Data are presented for three independent 15 mL batch cultures (n = 3). *P* values were obtained from one-way ANOVA (Dunnett's test) (GraphPad Prism 9) by comparing 100% with decreasing concentrations of ASW.
(PDF)

**S5 Table. Values behind means, SEM, and *p*-values used in Fig 1c.** Dry cell yield (g $L^{-1}$) of *R. sulfidophilum* in ASW supplemented with 1 mM sodium acetate and 2.5 mM sodium thiosulfate in decreasing concentrations of ASW, i.e., 100%, 90%, 80%, 70%, 60%, and 50% which correspond to 3%, 2.7%, 2.4%, 2.1%, 1.8%, and 1.5% salinities respectively (Fig 1c). Data are presented for three independent 15 mL batch cultures (n = 3). *P* values were obtained from one-way ANOVA (Dunnett's test) (GraphPad Prism 9) by comparing 100% with decreasing concentrations of ASW.
(PDF)

**S6 Table. Values behind means, SEM, and *p*-values used in Fig 2b.** Dry cell yield (g $L^{-1}$) of *R. sulfidophilum* in ASW supplemented with 0.1% yeast extract and 0.5% peptone at 100%, 50%, 40%, and 30% ASW, corresponding to 3%, 1.5%, 1.2%, and 0.9% salinities, respectively. (Fig 2b). Data are presented for four independent 10 L batch cultures (n = 4). *P* values were obtained from one-way ANOVA (Dunnett's test) (GraphPad Prism 9) by comparing 100% with decreasing concentrations of ASW.
(PDF)

**S7 Table. Values behind means, SEM, and *p*-values used in Fig 2c.** Initial and final OD at 660 nm for 100% and 40% ASW treatments at 10 L scale (Fig 2c). Data are presented for three independent 10 L batch cultures (n = 3). *P* values were obtained using Student's T-test statistic (Microsoft Excel 2019) by comparing 100% and 40% ASW treatments at the start of culture and the day of harvest.
(PDF)

**S8 Table. Values behind means, SEM, and *p*-values used in Fig 2d.** Dry cell yield (g $L^{-1}$) of *R. sulfidophilum* in 100% and 40% ASW treatments at 10 L scale (Fig 2d). Data are presented for three independent 10 L batch cultures (n = 3). *P* values were obtained using Student's T-test statistic (Microsoft Excel 2019) by comparing 100% and 40% ASW treatments at the start of culture and the day of harvest.
(PDF)

**S9 Table. Values used in Figs 3b and 3d.** Growth parameters of *R. sulfidophilum* in 100% or 40% ASW supplemented with 10 mM sodium thiosulfate pentahydrate and $CO_2/N_2$ (7:3) gas mixture (Figs 3b and 3d). Technical replicates are not available for these measurements.
(PDF)

**S10 Table. Values behind means, SEM, and *p*-values used in Fig 3c.** Dry cell yield (g $L^{-1}$) of *R. sulfidophilum* in 100% or 40% ASW supplemented with 10 mM sodium thiosulfate pentahydrate and $CO_2/N_2$ (7:3) gas mixture (Fig 3c). Data are presented for a single 2 L batch culture with 3 technical replicates (n = 3). *P* values were obtained using Student's T-test statistic (Microsoft Excel 2019) by comparing 100% and 40% ASW treatments at corresponding time points. n.d. indicates no data.
(PDF)

**S11 Table. Values behind means, SEM, and *p*-values used in Figs 3e and 3f.** Total organic carbon (TOC) and total nitrogen (TN) (g L$^{-1}$) in *R. sulfidophilum* cells in 100% or 40% ASW supplemented with 10 mM sodium thiosulfate penta-hydrate and $CO_2/N_2$ (7:3) gas mixture (Figs 3e and 3f). Data are presented for a single 2 L batch culture with 3 technical replicates (n = 3). *P* values were obtained using Student's T-test statistic (Microsoft Excel 2019) by comparing 100% and 40% ASW treatments at corresponding time points.
(PDF)

## Acknowledgments

We would like to thank Dr. Nehlah Rosli (Symbiobe Inc.) for her assistance with preculture preparation for the autotrophic cultivation of *R. sulfidophilum.*

## Author contributions

**Conceptualization:** Shamitha Rao Morey-Yagi, Keiji Numata.

**Data curation:** Shamitha Rao Morey-Yagi, Dao Duy Hanh, Hiromasa Morishita.

**Formal analysis:** Shamitha Rao Morey-Yagi, Hiromasa Morishita.

**Funding acquisition:** Keiji Numata.

**Investigation:** Shamitha Rao Morey-Yagi, Dao Duy Hanh, Miki Suzuki, Shota Kato, Geoffrey Liou, Yuki Kuroishikawa, Ayaka Yamaguchi, Hiromasa Morishita, Masaki Odahara, Keiji Numata.

**Methodology:** Shamitha Rao Morey-Yagi.

**Project administration:** Keiji Numata.

**Resources:** Shamitha Rao Morey-Yagi.

**Supervision:** Keiji Numata.

**Validation:** Shamitha Rao Morey-Yagi, Keiji Numata.

**Visualization:** Shamitha Rao Morey-Yagi.

**Writing – original draft:** Shamitha Rao Morey-Yagi, Hiromasa Morishita.

**Writing – review & editing:** Keiji Numata.

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
