## [Decision Letter · Decision Letter 0]

Dear Dr. Numata,

Thank you for submitting your manuscript to PLOS ONE. After careful consideration, we feel that it has merit but does not fully meet PLOS ONE’s publication criteria as it currently stands. Therefore, we invite you to submit a revised version of the manuscript that addresses the points raised during the review process.

We look forward to receiving your revised manuscript.

Kind regards,

Ashfaq Ahmad, Ph.D

Academic Editor

PLOS ONE

Journal Requirements:

Numata K., Morey-Yagi SR., and Symbiobe Inc. are co-inventors of a patent for fertilizer produced using R. sulfidophilum biomass (Patent pending JP2022-076662). Kato S., Liou G., Kuroishikawa Y., Yamaguchi, A., and Numata K. have affiliations with Symbiobe Inc.

This work was supported by Japan Science and Technology, COI-Next (Grant Number JPMJPF2114).

We would like to thank Dr. Nehlah Rosli (Symbiobe Inc.) for her assistance with preculture preparation used in the autotrophic cultivation of R. sulfidophilum. This work was supported by Japan Science and Technology, COI-Next (Grant Number JPMJPF2114). The funder played no role in the study design, data collection, analysis, interpretation, or writing of this manuscript.

This work was supported by Japan Science and Technology, COI-Next (Grant Number JPMJPF2114).

6. We note that your Data Availability Statement is currently as follows: All relevant data are within the manuscript and its Supporting Information files.

7. Please amend your list of authors on the manuscript to ensure that each author is linked to an affiliation. Authors’ affiliations should reflect the institution where the work was done (if authors moved subsequently, you can also list the new affiliation stating “current affiliation:….” as necessary).

Reviewers' comments:

Reviewer's Responses to Questions

**Comments to the Author**

1. Is the manuscript technically sound, and do the data support the conclusions?

Reviewer #1: Yes

Reviewer #2: Yes

Reviewer #3: Yes

2. Has the statistical analysis been performed appropriately and rigorously?

Reviewer #1: No

Reviewer #2: Yes

Reviewer #3: Yes

3. Have the authors made all data underlying the findings in their manuscript fully available?

Reviewer #1: No

Reviewer #2: Yes

Reviewer #3: Yes

4. Is the manuscript presented in an intelligible fashion and written in standard English?

Reviewer #1: Yes

Reviewer #2: Yes

Reviewer #3: Yes

Reviewer #1: The authors of this study explore a cost-effective alternative to nutrient-rich culture media for large-scale biomass production of the marine purple photosynthetic bacterium Rhodovulum sulfidophilum, which is valuable in bioremediation, biotechnology, and agriculture. The researchers found that reducing salinity in seawater-based cultures from 3% to 1.2% did not negatively impact heterotrophic growth, cell yield, nitrogen content, or amino acid composition. Lower salinity also proved feasible for cultivation without raising contamination risks, offering a viable option for large-scale use as a plant nitrogen fertilizer. However, there are some concerns, as listed below:

a) The authors should study the long-term stability and viability of R. sulfidophilum cultures at reduced salinity levels.

b) The authors should comment on the environmental impact of using low-salinity seawater for large-scale cultivation.

c) The authors should provide a zoom in picture of Fig1A to clearly distinguish the difference of OD600 at 175 hours of culture

d) The authors should confirm the number of trials, especially for 60 and 50% ASW, in Fig 1b

e) Figs. 2 and 3 need clarification on statistics, and the authors should provide all the raw data and statistics. Fig 2c,d need statistics and Fig 3e requires further explanation of which culture performed better.

Reviewer #2: Strengths:

• Clearly defined experimental setup across heterotrophic and autotrophic conditions.

• Comprehensive detail on salinity conditions, growth curve, biomass harvesting, and chemical analysis.

• Inclusion of contamination check and sterilization methods shows good microbiological practice.

• The use of seawater and cost-effective flocculation steps shows practical relevance.

Some lacks:

1. Does the study present actual plant growth trails under greenhouse or field conditions using the biomass obtained from 1.2%?

2. The author mentioned “cost effective”. So, the author should provide details.

3. Does the study assess the long-term environmental impact or biodegradability of the biomass when used as fertilizer?

4. Lysed biomass as fertilizer (lines 14–15) and again in lines 16–18 feels slightly repetitive.

5. The "gap" could be emphasized more for a strong impact. If it is possible.

6. Citations (e.g., 1–18) are referenced extensively, but some seem clustered without specifics. It's unclear which statements each citation supports.

7. Line 161: “R. sulfidophilum (DSM 1374; W4, LMG 5202) (ATCC)” is slightly confusing.

8. Suggestion: Clarify the exact strain and source used:

“Rhodovulum sulfidophilum (DSM 1374, equivalent to LMG 5202; obtained from ATCC) was used for all cultivation experiments.”

If multiple strains or IDs are applicable, clarify which is the principal one.

Reviewer #3: Peer Review Report

Recommendation: Minor Revision

This manuscript presents a well-structured and carefully executed study that investigates the potential for using a low-salinity artificial seawater medium for the large-scale cultivation of Rhodovulum sulfidophilum. The work is highly relevant to sustainable biotechnology, particularly for cost-effective biomass production with applications in biofertilizers.

Q1. Technical Soundness and Support for Conclusions

Yes. The experiments were thoughtfully designed and appropriately replicated (e.g., four biological replicates for 10 L batch cultures), with robust comparisons across conventional marine broth and both natural and artificial seawater-based media. The core findings—that a salinity reduction from 3% to 1.2% does not compromise growth, biomass yield, or biomass quality—are well supported by both qualitative observations and quantitative data. The authors also checked for contamination and demonstrated that low salinity did not increase contamination risk. Overall, the conclusions are well grounded in the evidence provided.

Q2. Statistical Analysis

Yes. The manuscript demonstrates sound use of statistics. Two-way ANOVA with post-hoc Tukey tests is used appropriately for comparisons across salinities and media types, while Student’s t-tests are applied to pairwise comparisons. The authors provide mean values and SEM, and statistical significance is clearly indicated throughout the figures. One suggestion would be to clarify in the text whether phrases like “tended to be higher” refer to statistically significant differences or trends that did not reach significance.

Q3. Data Availability

Yes. The authors clearly state that all data underlying the results are included in the manuscript and supporting information files. Key results are well-documented in figures and tables, with no apparent omissions. The study complies with PLOS ONE’s data availability policies.

Q4. Clarity of Presentation

Yes. The manuscript is generally well-written, with clear and logical flow. The introduction provides adequate context and rationale, and the results and discussion are well organized. While the English is fluent and professional, a light proofreading for minor typographical inconsistencies would be beneficial. For example, compound words like “cost-effective” and unit formatting (e.g., “g L⁻¹” vs “g L-1”) should be made consistent throughout the manuscript.

Review Comments to the Authors

General Assessment:

This study is a valuable contribution to microbial biotechnology and sustainable agriculture. It offers a practical solution to reduce media cost in large-scale microbial cultivation without sacrificing performance or introducing new risks. The use of both heterotrophic and autotrophic growth conditions strengthens the applicability of the findings.

Strengths:

• The stepwise experimental design—starting with media comparison and moving toward salinity optimization—is a strong feature.

• The cost analysis is timely and relevant for scaling microbial production.

• The consistent performance of the 1.2% salinity medium under both heterotrophic and autotrophic conditions is clearly demonstrated.

• Contamination checks, amino acid profiling, and N:P:K analysis add meaningful depth to the study.

•

Suggestions for Improvement:

1. Statistical clarity: Please indicate in the text whether observed differences (e.g., higher N content or amino acid levels at 1.2% salinity) are statistically significant or simply trends. If tests were not applied to pooled amino acid data, a note to that effect would be helpful.

2. Table 1 presentation: Clarify how “relative dry cell yield” was calculated and consider revising the phrasing around the cost difference to make it more intuitive.

3. Free amino acid profiles: Since Table 2 presents data from pooled samples, please indicate that error bars are not available and explain briefly how this affects interpretation.

4. Terminology: In the Abstract or Introduction, consider stating that 3% NaCl is equivalent to the salinity of natural seawater. Readers from outside marine microbiology may not know this.

5. Formatting and style: Ensure consistent formatting for units and superscripts in figure legends (e.g., “10⁻⁶” rather than “10-6”).

6. Figure legends: Please make sure each figure legend explains the basis of statistical comparisons—e.g., whether comparisons are across all salinities or between two specific groups.

7. Literature integration: References [13] and [18] are cited several times. Consider acknowledging their foundational role in the discussion, especially in relation to media design and previous applications of R. sulfidophilum biomass as fertilizer.

**Do you want your identity to be public for this peer review?** For information about this choice, including consent withdrawal, please see our Privacy Policy

Reviewer #1: No

Reviewer #2: No

Reviewer #3: No

---

## [Author Response · Author response to Decision Letter 1]

15 May 2025

the responses are attached as a separated file. thank you

---

## [Editor Report · Decision Letter 1]

Low-salinity medium for large-scale biomass production of the marine purple photosynthetic bacterium Rhodovulum sulfidophilum

PONE-D-25-12993R1

Dear Dr. Numata,

We’re pleased to inform you that your manuscript has been judged scientifically suitable for publication and will be formally accepted for publication once it meets all outstanding technical requirements.

Kind regards,

Ashfaq Ahmad, Ph.D

Academic Editor

PLOS ONE

---

## [Editor Report · Acceptance letter]

PONE-D-25-12993R1

PLOS ONE

Dear Dr. Numata,

I'm pleased to inform you that your manuscript has been deemed suitable for publication in PLOS ONE. Congratulations! Your manuscript is now being handed over to our production team.

Kind regards,

on behalf of

Dr. Ashfaq Ahmad

Academic Editor

PLOS ONE